# Land use and pollinator dependency drives global patterns of pollen limitation in the Anthropocene

Joanne M. Bennett [1,2,3 ✉], Janette A. Steets[4,5], Jean H. Burns [6], Laura A. Burkle [7], Jana C. Vamosi [8], Marina Wolowski[9], Gerardo Arceo-Gómez[10], Martin Burd [11], Walter Durka [12], Allan G. Ellis[13], Leandro Freitas [14], Junmin Li[15], James G. Rodger[13,16,17], Valentin Ştefan[2,12], Jing Xia[18], Tiffany M. Knight [1,2,12,20] & Tia-Lynn Ashman [19,20]

Land use change, by disrupting the co-evolved interactions between plants and their pollinators, could be causing plant reproduction to be limited by pollen supply. Using a phylogenetically controlled meta-analysis on over 2200 experimental studies and more than 1200 wild plants, we ask if land use intensification is causing plant reproduction to be pollen limited at global scales. Here we report that plants reliant on pollinators in urban settings are more pollen limited than similarly pollinator-reliant plants in other landscapes. Plants functionally specialized on bee pollinators are more pollen limited in natural than managed vegetation, but the reverse is true for plants pollinated exclusively by a non-bee functional group or those pollinated by multiple functional groups. Plants ecologically specialized on a single pollinator taxon were extremely pollen limited across land use types. These results suggest that while urbanization intensifies pollen limitation, ecologically and functionally specialized plants are at risk of pollen limitation across land use categories.

[1] Institute of Biology, Martin Luther University Halle-Wittenberg, Am Kirchtor 1, 06108 Halle (Saale), Germany. [2] German Centre for Integrative Biodiversity Research (iDiv) Halle-Jena-Leipzig, Deutscher Platz 5e, 04103 Leipzig, Germany. [3] Centre for Applied Water Science, Institute for Applied Ecology, Faculty of Science and Technology, University of Canberra, Canberra, Australia. [4] Department of Plant Biology, Ecology and Evolution, Oklahoma State University, Stillwater, OK, USA. [5] Illumination Works, 2689 Commons Blvd, Suite 120, Beavercreek, OH 45431, USA. [6] Department of Biology, Case Western Reserve University Cleveland, Ohio 44106-7080, USA. [7] Department of Ecology, Montana State University, Bozeman, MT 59717, USA. [8] Department of Biological Sciences, University of Calgary, Calgary, AB, Canada. [9] Institute of Natural Sciences, Federal University of Alfenas, Alfenas, Brazil. [10] Department of Biological Sciences, East Tennessee State University, Johnson City, TN, USA. [11] School of Biological Sciences, Monash University, Melbourne, Australia. [12] Department of Community Ecology, Helmholtz Centre for Environmental Research—UFZ, Theodor-Lieser-Straße 4, 06120 Halle(Saale), Germany. [13] Department of Botany and Zoology, University of Stellenbosch, Private Bag X1, Matieland 7602, South Africa. [14] Rio de Janeiro Botanical Garden, Rio de Janeiro, Brazil. [15] Taizhou University, Jiaojiang District, Taizhou City, Zhejiang, P. R. China. [16] Biodiversity Informatics Unit, Department of Mathematical Sciences, Stellenbosch University, Private Bag X1, Matieland 7602, South Africa. [17] Department of Ecology and Genetics, Uppsala University, Uppsala, Sweden. [18] College of Life Sciences, South-Central University for Nationalities, Wuhan, Hubei, P. R. China. [19] Department of Biological Sciences, University of Pittsburgh, Pittsburgh, PA 15260, USA. [20] These authors contributed equally: Tiffany M. Knight, Tia-Lynn Ashman. ✉email: joanne.bennett@canberra.edu.au

Nearly 90% of flowering plants rely on animal pollinators for reproduction[1], and as a consequence, angiosperm biodiversity relies on stable mutualisms between plants and pollinators[2,3]. As the world's human population has grown, native vegetation has been converted to intensively human-managed and urbanized landscapes[4] that, along with increased use of pesticides, have demonstrably reduced pollinator abundance and diversity even in natural areas[4–8]. Although insect declines are now recognized broadly, wild bee species may be particularly vulnerable to land-use change[9,10] and these represent the most important pollinators of flowering plants globally[5,11]. Moreover, how plant reproduction responds to land use via any declines in pollinators has important implications for much of the world's flora[12], yet the effects of land use changes on pollen limitation of wild plant reproduction have not been evaluated on a global scale[13].

The consequences of anthropogenic disturbances for pollen limitation of plant reproduction (hereafter PL) are likely to vary with degree of plant dependence on pollinators, as well as level of ecological or functional specialization[14], in addition to plant traits that reflect the evolutionary history of their interactions with their pollinators, such as floral symmetry[15,16]. For example, plant species that have evolved traits that buffer them from pollinator uncertainty, such as autofertility (i.e., self pollination in the absence of flower visitors) and functional generalization (e.g., pollination by a range of taxa or functional groups), are expected to be less prone to PL in response to anthropogenic change. While land use changes have been posited to erode ecosystem services provided by pollination, the effects of land use change on plants is likely heavily mediated by pollinator dependence. Thus, the consequences of land use change on PL and on how it may reshape phenotypic and genetic diversity, as well as the distributions of plant species across the globe require a more nuanced examination.

The degree to which pollen receipt limits plant reproduction has been studied in thousands of independent experiments that compare fruit or seed production of flowers exposed to natural pollination with those receiving supplemental pollination. This standardized experimental approach provides important insight to assess global drivers of PL via meta-analysis while controlling for plant phylogenetic history[17,18]. Early theoretical research based on sexual selection and optimality predict that plants should not increase seed production in response to experimental pollen addition unless they have been displaced from their evolutionary optimum[16,19–21], possibly by anthropogenic factors. While later models have suggested that PL may represent an evolutionary equilibrium in a stochastic pollination environment where pollen quantity or quality may vary[19,22,23], anthropogenic changes that disrupt plant–pollinator interactions beyond historical means and variances are still expected to increase PL. Yet we do not know the extent of anthropogenic impact nor the spatial scale at which it occurs.

In this study, we use phylogenetically controlled meta-analysis of 2247 studies of 1247 wild plant species across the globe (Fig. 1a) in conjunction with data on landscape conversion to determine whether there is a signature of contemporary land use on PL, and if so, whether it is dependent on the extent to which plant species rely on pollinators for reproductive success. Does high pollinator dependency and high ecological or functional pollinator specialization place plants at higher risk of PL, while autofertility or pollinator generalization buffer plant reproduction from PL, in the face of land use modification?

We show that pollinator dependant plants in urban settings have higher PL than those in managed and natural landscapes, and that plant traits play a strong role in determining PL across different land use categories. Our results show that high intensity land-use increases PL, and that ecologically and functionally specialized plants are particularly at risk. This work reveals that human-mediated disruptions may be a turning point for natural systems, and that conservation should focus not just on pollinators but also the diverse wild plant communities that support them, especially in urban and natural habitats.

## Results

**Global patterns in PL**. PL was evident at a global scale: on average the PL effect size in GloPL[17] is 0.49 (CI: 0.45–0.52), which equates to a 63% increase in reproduction following supplementation (Fig. 1b). We did not find significant phylogenetic signal in PL in our highly geographically and species diverse dataset ($K = 0.31$, $P = 0.097$). However, as a variety of plant traits related to pollination have been shown to be phylogenetically conserved[24,25], we control for phylogenetic structure in the meta-analysis and focus on the influence of land use categories and pollinator dependency on PL. Land use categories, pollinator dependency, ecological specialization and functional specialization in our data set were well distributed across the globe (Fig. 1a) and across our plant phylogeny (Fig. 2a).

**Land-use intensity**. The effects of land use on PL were influenced by pollinator dependency (Supplementary Tables 1 and 2; Fig. 1b —$Q_M = 13{,}294$, d$f = 6$, $P < 0.001$). Autofertile plants were not PL under any land use category (none significantly different from zero, Fig. 1b, Supplementary Table 1). However, for pollinator-dependent plants, the extent of PL depended on land use with PL greatest in urban locations, followed by natural and managed vegetation (Fig. 1b; Supplementary Tables 1 and 2). Although the frequency of studies in urban landscapes is low, the result is robust and is derived from 93 studies conducted in 24 urban centers across the globe (Fig. 1a).

**Ecological and functional specialization**. Plants only pollinated by one pollinator taxon have higher PL than those pollinated by few or many pollinator taxa (Supplementary Table 3; Fig. 2a). Functional specialization significantly modified responses of PL to land use (Supplementary Tables 4 and 5—$Q_M = 4518$, d$f = 6$, $P < 0.001$). Specifically, exclusively bee-pollinated plants were significantly more PL in natural landscaped than in managed landscapes (Fig. 2c, Supplementary Table 5), but the opposite was the case for plants exclusively pollinated by another functional group or those serviced by multiple functional groups. For these, managed habitats lead to higher PL than natural ones (Fig. 2c, Supplementary Table 5).

## Discussion

Our finding of higher PL in urban settings suggests that urbanization (e.g., fragmentation, impervious surfaces, and pollution and traffic) is highly disruptive to plant–pollinator interactions[26]. This result reflects recent reports suggesting that although pollinator richness can be high in urban areas, pollinators tend to service a lower proportion of the available plant species than in managed and natural sites[27]. Plants in managed and natural habitats are similarly pollen limited (Table S1; Fig. 1b). Variation in intensity of management and/or in degree of degradation of natural habitats could be obscuring potential differences in these land use categories, or it is possible that differences in PL depend on ecological and functional specialization on pollinators. For example, although many stressors associated with managed landscapes are known to lead to higher PL[14], heterogenously managed landscapes can also increase pollinator diversity and therefore could lower PL[10]. Furthermore, the asymmetric nature of plant–pollinator interactions, in which specialist plant species

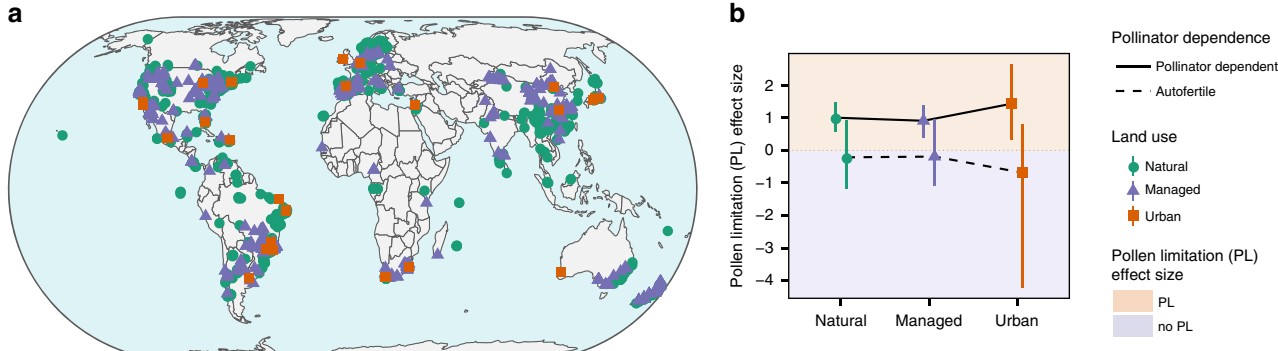

**Fig. 1 The global distribution of data from the GloPL database (a) and an interaction plot showing the interaction between land use and pollinator dependence in respect to the effect size of pollen limitation (PL) (b).** The point colour indicates the dominant land use category urban (orange), managed (purple), and natural (green) in (**a**, **b**). In the interaction plot, pollinator dependant plants are indicated by the solid line and autofertile plants by the dashed line. The area of the plot shaded orange indicates an effect size above (i.e. plants are PL) and the area of the plot shaded purple indicates an effect size below (i.e. plants are not PL). The interaction plot illustrates the average modelled result and 95% confidence intervals (shown as error bars) from 500 bootstrapped phylogenetic meta-analyses with the response variable PL and the interaction between land use and pollinator dependence as the predictor variables. Source data are provided as a Source Data file.

are often pollinated by generalist pollinators, may make them resilient to some disturbance[28].

In both managed and natural landscape types, we found that the most ecologically specialized plants—those pollinated by only one pollinator taxon—were generally more pollen limited than those pollinated by few or many pollinator taxa (Supplementary Table 3; Fig. 2a). These results indicate that regardless of contemporary land use, reproduction by highly specialized plant species, such as orchids, and endangered endemic species, such as *Daphne rodriguezii* (Thymelaeaceae) and *Oxypetalum mexiae* (Apocynaceae), are vulnerable to pollinator declines at a global scale.

While insects are declining globally[5], losses are not uniform across taxa and habitat types[29], and the composition and efficiencies of pollinator fauna can differ among habitat types[30]. For example, in the UK, rare bee species have strongly declined in natural habitats, while widespread generalist bees (that are dominant crop pollinators) have increased in managed habitats[29]. In contrast to native pollinators, global trends suggest managed honey bee hives are increasing[31]. In many managed habitats, pollination is supplemented by domesticated honey bees, and this could lower PL not only for the crop species but also for the wild plants in these settings[32]. However, the addition of honey bees can have detrimental effects on other pollinating taxa, negatively impacting the plant species that rely on them[33]. We expected that plants exclusively pollinated by bees might benefit from managed habitats while those specialized on other functional groups (e.g., dipterans, lepidopterans, and mammals) might not. We expected that plants pollinated by multiple functional groups including bees (e.g., species visited by two or more orders of insects) would have low levels of PL across both land use types. We find that exclusively bee-pollinated plants were significantly more PL in natural habitats than managed ones (Fig. 2c; Supplementary Table 5), but the opposite was the case for plants exclusively pollinated by another functional group or those serviced by multiple functional groups. For these, managed habitats lead to higher PL than natural ones (Fig. 2c; Supplementary Table 5). The result of enhanced reproductive output of bee-pollinated plant species in managed areas is consistent with the findings that bee abundance is also higher in managed areas[34], thereby highlighting how understanding the pollinator crisis requires more research effort on non-bee pollinators and non-bee pollinated plant species. Taken together these results highlight the complex ways that land use intensification along with other anthropogenic forces put various wild plant species at risk of reproductive failure.

On a global scale, we found that PL was related to the intensity of human land use and that the magnitude of the effect was modulated by plant traits that reflect their dependence and specialization on pollinators. Our results link anthropogenic disturbance and changes in pollinator services to plant reproduction and, by doing so, fill a major gap in our knowledge highlighted in the recent Intergovernmental Science-Policy Platform on Biodiversity and Ecosystem Services Pollinators, Pollination and Food Production assessment[11]. The magnitude of PL in pollinator-dependent plants in natural sites highlights that to maintain healthy plant communities under widespread pollinator declines new management approaches that incorporate natural landscapes are needed. This is particularly urgent because pollinator losses may set in motion negative feedback loops where loss of pollinators limits reproduction which leads to plant population declines that lead to even greater pollinator declines. This may occur even for pollinators that are more resistant to anthropogenic change, e.g., generalist crop pollinators, as even these need diverse plant communities for temporal stability and diversity in floral resources, as well as diverse nesting habitat[5,6]. In the longer term, evolution toward autofertility and/or pollination generalization[35] could buffer many plant species from pollinator losses. However, evolution towards increasing reliance on generalist pollinators could result in a dead end if pollinator losses continue. On the other hand, evolution toward selfing can decrease overall genetic diversity leaving plants vulnerable to extinction under further environmental perturbation[35]. Species that self pollinate also allocate less to pollen and nectar, than outcrossing species, additionally reducing resource availability to pollinators[36]. Recognizing that human-mediated disruptions may represent a turning point for these natural systems, conservation should focus not just on pollinators but also the diverse wild plant communities that support them, especially in urban and natural habitats.

## Methods

**Experimental design.** We used data from 2247 study populations of 1247 plant species across the globe from the GloPL database[17]. Each experiment compared the mean reproductive output of plants receiving supplemental pollination applied by hand with those receiving natural pollination. A pollen limitation effect size was calculated as the log response ratio of reproduction following natural or supplemental pollination[2,3]: PL effect size = ln [(supplement)/(natural)]. The response variables (i.e., reproductive output in natural or supplemental flowers) were based on one of fruit set, seed set, seeds per fruit, seeds per flower, or seeds per plant. We computed a single estimate of the magnitude of PL and its variance for each unmanipulated experiment (i.e., species, population, and year of study). In simple cases, a pooled variance was calculated following ref. [37], page 64, i.e., when a row

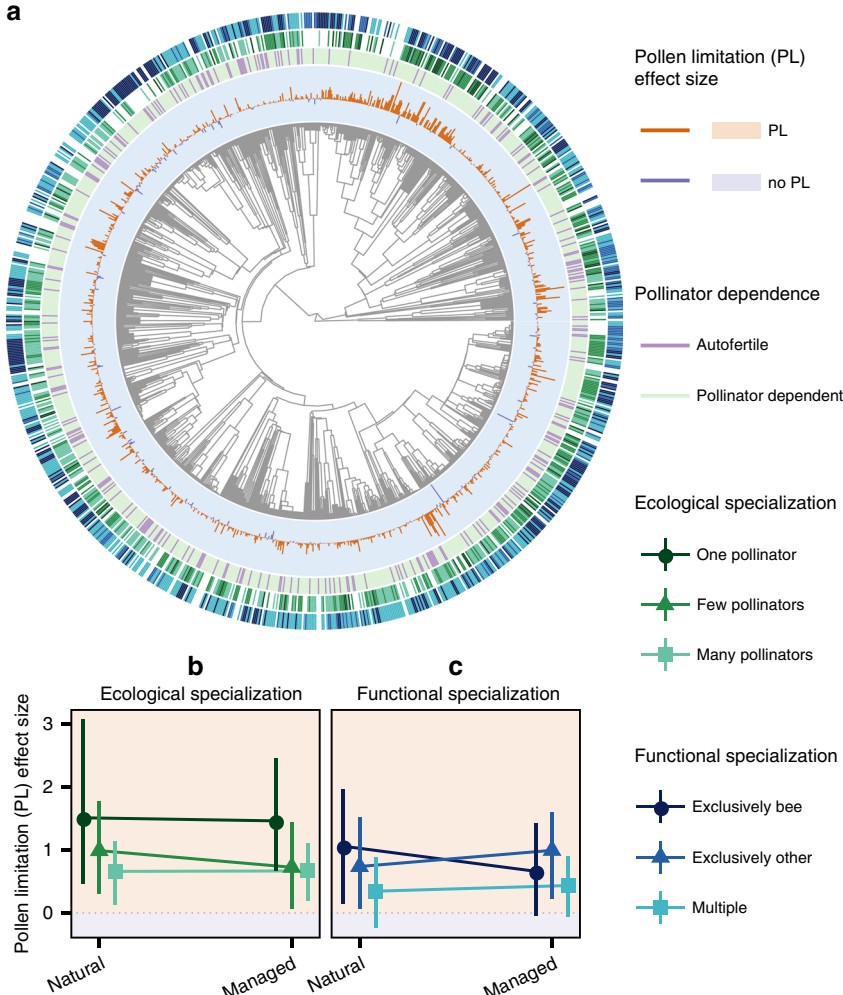

**Fig. 2 Phylogenetic distribution of data extracted from the GloPL database[17] (a) and interaction plots of the interaction between land use and ecological specialization (b) and land use and functional specialization (c) in respect to the effect size of pollen limitation (PL).** The phylogeny is modified from the angiosperm supertree[42] and for each species the PL effect size and category of pollinator dependence, ecological specialization, and functional specialization are shown. Pollen limitation effect size in (**a**) is given by a bar plot, where orange bars indicate a positive effect size and dark purple bars indicate an effect size of or below (i.e. no PL). Pollinator dependence of plants in (**a**) is classified as autofertile (purple) or pollinator dependent (light green). Ecological specialization of plants in (**a**, **b**) is classified as reliant on either one (dark green), few (green) or many (light green) pollinator species. Functional specialization of plants in (**a**, **c**) is classified as exclusively bee pollinated (dark blue), exclusively pollinated by another functional group (blue) or pollinated multiple functional groups (light blue). Interaction plots represent the average modelled and 95% confidence intervals (shown as error bars) result from 500 bootstrapped phylogenetic meta-analyses with pollen limitation as the response variable and the interaction between land use and ecological specialization or functional specialization as the predictor variables. Source data are provided as a Source Data file.

related to a single species population and year. For cases in GloPL when data for a single species were presented across multiple rows because there were multiple time-periods (e.g., season) or multiple morphs (e.g., flower color and gender) variance was calculated following ref. [38] formulae 11.2–11.4, pages 65–66. A small value was added to all cases so that zero cases could be included in the calculations of variance. We compared results with this PL effect size to those where 0.5 was added to both the response variables before log transformation, in cases where one or both of the response variables was zero. This leads to a slightly larger sample size (~2% increase), because points with zero values (e.g., no seed set under natural conditions) can be included. Analysis using both response variables were the same and the interpretations unaffected, therefore we only present model results from the more conservative PL effect size with zero values excluded.

**Land use variables**. We used the spatial coordinates supplied in the GloPL dataset[17] to determine land use. Land use percent cover in 12 categories urban, agricultural crops (5 categories; C3 nitrogen fixing, annual and perennial and C4 annual and perennial), rangeland, pasture, primary forest, primary non-forest, secondary forest, and secondary non-forest was extracted using the GPS location and the year of study from the Land-Use Harmonization 2 (LUH2) dataset[39] which contains annual land use states for the years 850–2100 at 0.25° × 0.25° spatial

resolution. The dominant land use category surrounding each PL experiment was consolidated to three main category types: 'urban', 'managed' (agricultural crops, rangeland, and pasture), 'natural' vegetation (primary and secondary forest or non-forest). In the LUH2 dataset[39] the rangeland classification is based on the aridity index and the human population density and could range from semi-natural vegetation grazed by livestock to intensively managed pastures, e.g., were broadleaf herbicide are applied to reduce non-grass species. For this reason, we performed analyses both with and without rangelands included in the 'managed' category but found no difference in the quantitative results, thus we retained rangeland in the managed category presented here. We acknowledge that the broad categories of land-use used here are unlikely to capture the full range of intensity of urban, managed or natural environments. However, there are clear advantages to using such broad categories of land-use. Firstly the data is available at a global scale and secondly these broad categories are relevant to all biogeographic regions. Given the large numbers of species and the vast geographic area of coverage, this leads to the expectation that general patterns should still emerge, if present.

**Pollinator dependency traits**. Plants were scored as pollinator dependent when evidence of pollinator dependence existed, that is they were reported to be pollinator dependent, self-incompatible, or self-compatible but not autofertile following[24].

When quantitative data was not available, we scored the trait based on the author's statements first and then considered information from additional published literature and web sources. Diecious, distylous and tristylous species were categorized as pollinator dependent. Information on pollinator dependency status was missing for 60 records, these along with wind-pollinated plants were excluded from analysis.

Levels of pollination specialization were scored based on the authors determination in the original studies. The degree of ecological specialization was based on the total number of known pollinators for the plant or the number of recorded flower visitors to the plant. Plants were scored as 'one' when pollinated by one pollinator species, 'few' when pollinated by a few (2–4) species or 'many' (5 or more) pollinator species following[25]. The degree of functional specialization was characterized as 'exclusively bee', when pollinated by this functional group, the largest and most efficient pollinating class[10] and the majority of functionally specialized plants in our dataset, or as 'exclusively other' when pollinated by a single other functional group (i.e., either flies, beetles, moths, butterflies, wasps, mammals, or birds) or as 'multiple groups' when pollinated by multiple functional groups, including bees and others. As with all meta-analysis there will be sampling differences between studies and these may affect our measures of ecological and functional specialization. However, the authors of each study are assumed to be the authority on their study species and we do not expect bias to occur in any particular direction. Thus, given the large sample size of our dataset broad patterns should still emerge if present.

**Statistical analysis**. All analyses were performed in R version 3.6.3[40]. We conducted phylogenetic mixed-effects meta-analyses as per methods in refs. [24,41] with PL as the response variable and the interaction between land use, and three plant traits that relate to their level of dependence on pollinators (pollinator dependency, and ecological and functional specialization on pollinators). We used a phylogenetic meta-analysis, as in addition to weighting effect sizes by the inverse of their variances it incorporates a variance-covariance structure based on phylogenetic relationships to take the non-independence among species into account[18]. The species-level phylogeny used in our analysis is available on-line as part of the GloPL database[17].

To create the phylogeny, we started with the dated supertree created by Zanne et al.[42]. Species that were not included in the supertree, were bound to the tree when their genus was present by creating a polytomy with congeners that were present in the tree using the congeneric.merge function from the 'pez' package in R[43]. When no congener was present, as was the case for 60 of the GloPL species, we searched the literature for published phylogenies indicating closely related genera and manually grafted these species to the branch leading to the genus clade. We then pruned the supertree to only include our focal species using the drop.tip function from the 'ape' package[44].

Phylogeny was modeled as a variance-covariance matrix, which assumes Brownian motion like evolution, using the vcv function in the ape package[44] and was included as a random effect in all models. Because differences in experimental design affect the estimated magnitude of PL, for a review of their effects see[45], we included in each model a random effect to control for differences in the response variables measured (fruit set, seed set, seeds per plant, seeds per flower, and seeds per fruit), the level at which the treatment was applied (whole plant, partial plant, and flower) and whether or not bags were applied to the plants. AIC model selection confirmed our strong a priori reasons for including all random and fixed effects used in each model. Overdispersion is common in meta-analysis and it is often necessary to include a random effect for each effect size Tau$^2$ as a correction. To test whether overdispersion is present and whether it affects our results we re-ran our models with the addition of a random effect for Tau$^2$. We found that our main result is robust to its inclusion and that none of our observed patterns changed (see Supplementary Tables 6–11). The rma.mv function in the metafor package version 2.4-0 was used to perform all models[46]. All models presented here were fit using ML and no quantitative differences were detected when compared with models fit using REML. To test for significant interactions between predictors we used the Holm adjustment for multiple comparisons[47] to conducted planned comparisons among means when appropriate. Profile plots of the variance component of each model was examined to insure there was a clear peak in likelihoods at the ML estimate, indicating the model had converged. Residuals were checked for normality and model fit.

For each figure presented in text we derived 95% confidence intervals around the model coefficients. We used a nonparametric bootstrap approach where each of our models was bootstrapped 500 times, sampling with replacement records from each interaction (each group/combination formed by the two fixed effects, i.e., land use and the three levels of dependence/specialization on pollinators). Marginal means for each group present in Fig. 1 were extracted by running bootstrapped models fit with ML without the intercept. Averaged bootstrapped model results are shown in text. All natural populations in GloPL with geographic coordinates, data on all random effect and with known pollinator dependency were included in modeled analysis.

**Reporting summary**. Further information on research design is available in the Nature Research Reporting Summary linked to this article.

## Data availability

The GloPL dataset is published in scientific data https://doi.org/10.1038/sdata.2018.249 and publicly available in the Dryad repository https://doi.org/10.5061/dryad.dt437. The

Land-Use Harmonization 2 (LUH2)[39] is publicly available here http://gsweb1vh2.umd.edu/LUH2/LUH2_v2h/states.nc. Source data are provided with this paper.

## Code availability

The associated analysis code and complementary functional and ecological data are archived on github (https://github.com/idiv-biodiversity/pollen-limitation-land-use).

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

## Acknowledgements
This paper is the result of working group sPLAT supported by sDiv, the Synthesis Center of the German Center for Integrative Biodiversity Research (iDiv) Halle-Jena-Leipzig (DFG FZT 118-202548816). Additional funding was provided by the Alexander von Humboldt Foundation as part of the Alexander von Humboldt Professorship of TMK, by the Helmholtz Association as part of the Helmholtz Recruitment Initiative to T.M.K. and the Helmholtz Association International Fellowship to T-L.A., and NSF (DEB1452386) to T-L.A. Early support was received as part of a Pollen Limitation Working Group supported by the National Center for Ecological Analysis and Synthesis, a Center funded by NSF (DEB-00,72909). We would like to thank the many authors of the original publications for their work. We thank S. Renner and the Munich Botanical Garden, Squire Valleevue Farm and Valley Ridge Farm at Case Western Reserve University, Janette and Michael Breese, K. Kietzmann, and N. Becker for logistical support. LF was supported by a CNPq PQ-grant.

## Author contributions
J.M.B. lead the analysis, data collection, and writing of the paper. J.A.S. conceived the project, contributed to the data collection, and edited the paper. J.H.B. contributed to the analysis and data collection and edited the paper. L.A.B. contributed to the data collection and edited the paper. J.C.V. contributed to the data collection and edited the paper. M.W. contributed to the data collection and edited the paper. G.A.-G. contributed to the data collection and edited the paper. M.B. contributed to the data collection and edited the paper. W.D. contributed to the data collection and edited the paper. A.G.E. contributed to the data collection and edited the paper. L.F. contributed to the data collection and edited the paper. J.L. contributed to the data collection and edited the paper. J.G.R. contributed to the data collection and edited the paper. V.Ş. contributed to analysis and figures and edited the paper. J.X. contributed to the data collection and edited the paper. T.M.K. conceived of the project and lead the writing of the paper. T-L.A. conceived the project and lead to the writing of the paper.

## Competing interests
The authors declare no competing interests.
