## [Peer Review File · Nature Communications]

Reviewers' Comments:

Reviewer #1:

Remarks to the Author:

My background is in meta-analysis, and I was asked to evaluate what the authors achieved in this manuscript. I like what they did, but some details are underreported, and below I outline these to improve the clarity of their meta-analysis.

1) More details on the effect size is needed (see Suppl. Page 1). Were both the numerator and denominator means (i.e. averages of supplemented and natural)? Please include the equation used to estimate the variance of the effect size (e.g., $\ln RR$). A clear description of the number of effect sizes included in the meta-analysis is missing, as well as the number of effects included in each subgroup analyses. The sample sizes of meta-analysis is defined with k and not N (since N is used to define the sample size used to estimate individual effect sizes). Please report all k (number of effect sizes) for each pooled effect size reported in the MS.

2) More information about the phylogeny used in modelling the meta-analysis is needed. Is this a species-level phylogeny? How were polytomies resolved? Surely not all species in the GloPI database were represented in the Zanne et al. phylogeny. Were species surrogates used for placement in the phylogeny? Perhaps all of this is all outlined in the GloPI publication, but some information is still needed here to make sense of the scope of this meta-analysis.

3) There are many random-effects included in the meta-analysis. The variances from these random-effects can swamp the weighting of effects, and therefore at minimum, all of the random-effect variances should be reported for at least the overall grand-mean synthesis (reporting this in the supplements is ok). This is important so that the reader can assess whether conventional between-study variance (τ -squared) assumed by random-effects meta-analysis is washed out by the numerous phylogenetic, response type, and experimental design random factors. It is also not clear whether the conventional τ -square was included in the meta-analysis; this is a critical random-effect that needs to be included in all models and underlies fundamental principles of meta-analysis. This is not included as a default when the `rma.mv` function is used in metafor. To include it, it is simply a column of unique numbers for each effect sizes used in the model.

4) About the meta-analysis model itself, what method was used to estimate the multiple random-effects (e.g., REML, ML)? Why was bootstrapping used? This seems unnecessary as the reported metafor coefficients and their 95%CI are sufficient (also 500 repetitions is very low given the number of effects included). This style of bootstrapping is also not part of the standard metafor functions of analyses; permutation tests are applied in separate functions, were these used? How were the marginal means of each group estimated when multiple predictors are included in models (e.g., results reported in Figures 1 and 2)? The estimated error type is not reported in either figure. It's not clear if interactions were included in models (e.g., land use type + pollinator dependence + land use type * pollinator dependence), or why it was excluded.

Reviewer #2:

Remarks to the Author:

Increasing pollination limitation is the expected consequence of pollinator decline, a by-product of man-driven conversion of natural habitats into extensive urban habitats and other types of land uses. The present MS "Land use and pollinator dependency drives global patterns of pollen limitation in the Anthropocene" addresses the question of whether land use intensification is associated with increasing pollen limitation by using a phylogenetically controlled meta-analysis based on a global dataset of over 2200 experimental studies and about 1200 wild plants. They found that urbanization as well as specialization increases pollination limitation.

Overall, this is an important, well-written and easy-to-read paper. I am quite certain that this study will be widely cited and have a high impact among policy makers and the general public. My main concerns are related to the model used for data analysis, as can be seen in the specific comments below. Also, I would like to see an extended discussion to explain the absence of a difference in pollen

limitation between natural and managed land. This is particularly puzzling given the finding that habitat fragmentation frequently leads to higher pollination limitation (Aguilar et al. 2006). This could relate to the fact that plants thriving in managed land have slightly different profiles in terms of pollination and mating systems than those surviving in the fragments, as it implied in the MS (lines 114-117). Other possibilities worth mentioning are that heterogenous managed landscapes can be pollinator rich (Winfree et al. 2007), or plant-pollinator interactions can be quite resilience due to their asymmetric nature (Ashworth et al. 2004). Also, it is interesting that specialized plants are the most vulnerable independent of habitat type, which is quite surprising as I might expect increasing pollen limitation in the most disturbed habitats. Beyond these issues, I am very positive about this article and with some extra clarification can be a very suitable contribution for Nature Communications.

Specific comments

Line 97-99. It should be interesting to know the magnitude of phylogenetic patterning (e.g., Bloomberg's K) existing in the data set.

Supplementary material

"We computed a single estimate of the magnitude of PL and its variance for each experiment (i.e., species, population, and year of study)." - PL is a transformed variable based on two input variables. The estimation of the variance of PL then needs clarification.

"Plants were scored as 'one' when pollinated by one pollinator species, 'few' when pollinated by a few (2 to 4) species or 'many' (5 or more) pollinator species following." - Most likely many studies report flower visitors rather than efficient pollinators. Therefore, clarification is needed here.

"All analyses were performed in R8. We conducted phylogenetic mixed effects meta-analyses as per methods in 5,9 with PL as the response variable and land use, levels of dependence on pollinators and their interaction as predictors"- What about the other fixed effects such as taxonomic and functional generalization? Also, there were more PL estimates than number of species. The presence of more than one PL estimate for some species needs clarification from a statistical point of views. Were estimates for single species averaged, or "species" included as a random factor?

"Phylogeny was modelled as a variance-covariance matrix, which assumes Brownian motion like evolution, using the vcv function in the ape package¹² and was included as a random effect in all models." - Phylogeny is an explanatory variable of seemingly random variation among species. I wouldn't consider phylogeny to be a random effect.

"Because differences in experimental design affect the estimated magnitude of PL, for a review of their effects see¹³, we included in each model a random effect to control for differences in the response variables measured (fruit-set, seed-set, seeds per plant, seeds per flower, seeds per fruit), the level at which the treatment was applied (whole plant, partial plant, flower) and whether or not bags were applied to the plants." - These are typical categories of fixed factors (i.e., "reproductive variable measured", "treatment level", and "bagging"); I am quite uncertain why they were considered random effects. Some clarification is needed.

Reviewers' comments:

Reviewer #1 (Remarks to the Author):

My background is in meta-analysis, and I was asked to evaluate what the authors achieved in this manuscript. I like what they did, but some details are underreported, and below I outline these to improve the clarity of their meta-analysis.

Response: We appreciate the endorsement and thankful for the advice on additional reporting metrics. We now provide a k for each subgroup in analysis in tables S1, S3 and S4. We have also included new tables S6-S11 which provide the estimate for each random effect included in the model.

1) More details on the effect size is needed (see Suppl. Page 1). Were both the numerator and denominator means (i.e. averages of supplemented and natural)? Please include the equation used to estimate the variance of the effect size (e.g., $\ln RR$). A clear description of the number of effect sizes included in the meta-analysis is missing, as well as the number of effects included in each subgroup analyses. The sample sizes of meta-analysis is defined with k and not N (since N is used to define the sample size used to estimate individual effect sizes). Please report all k (number of effect sizes) for each pooled effect size reported in the MS.

Response: Yes, both the supplemented and natural i.e. the numerator and denominator are means and we have revised the methods text on LN 4 to clarify this “Each experiment compared the mean reproductive output of plants receiving supplemental pollination applied by hand with those receiving natural pollination.”

Response: The sample size (k) is now reported for each subgroup in the analysis. Revisions made to tables S1, S3 & S4.

2) More information about the phylogeny used in modelling the meta-analysis is needed. Is this a species-level phylogeny? How were polytomies resolved? Surely not all species in the GloPI database were represented in the Zanne et al. phylogeny. Were species surrogates used for placement in the phylogeny? Perhaps all of this is all outlined in the GloPI publication, but some information is still needed here to make sense of the scope of this meta-analysis.

Response: Yes, the phylogeny is a species-level one and it was necessary to add some of our focal species to the tree. The methods text in LN 75 – 79 has been updated to clarify this point and now reads “The species-level phylogeny used in our analysis is available on-line as part of the GloPL database¹ and is based on the published dated molecular phylogeny by Zanne and colleagues¹¹. Focal species not present in the Zanne et al.¹³ phylogeny were bound on the tree when their genus was present, when not they were grafted on based on published information following methods outlined in GLoPL.”

3) There are many random-effects included in the meta-analysis. The variances from these random-effects can swamp the weighting of effects, and therefore at minimum, all of the

random-effect variances should be reported for at least the overall grand-mean synthesis (reporting this in the supplements is ok). This is important so that the reader can assess whether conventional between-study variance (τ -squared) assumed by random-effects meta-analysis is washed out by the numerous phylogenetic, response type, and experimental design random factors. It is also not clear whether the conventional τ -square was included in the meta-analysis; this is a critical random-effect that needs to be included in all models and underlies fundamental principles of meta-analysis. This is not included as a default when the `rma.mv` function is used in `metafor`. To include it, it is simply a column of unique numbers for each effect sizes used in the model.

Response: We have now included new tables S6-S11, which show the variance estimates from all random effects. We have also re-run the analysis to include a random effect for each estimate (τ^2). Including a τ^2 random effect did not change our main result that urban environments are more pollen limited and none of the patterns previously observed have changed.

4) About the meta-analysis model itself, what method was used to estimate the multiple random-effects (e.g., REML, ML)? Why was bootstrapping used? This seems unnecessary as the reported `metafor` coefficients and their 95%CI are sufficient (also 500 repetitions is very low given the number of effects included). This style of bootstrapping is also not part of the standard `metafor` functions of analyses; permutation tests are applied in separate functions, were these used? How were the marginal means of each group estimated when multiple predictors are included in models (e.g., results reported in Figures 1 and 2)? The estimated error type is not reported in either figure. It's not clear if interactions were included in models (e.g., land use type + pollinator dependence + land use type * pollinator dependence), or why it was excluded.

Response: All models were fitted using REML, this has been clarified in the methods text LN 93 which now reads "All models were fit using REML." Bootstrapping was used for the figures and not for model interpretation, this has now been clarified in the methods text L97-99, which reads "For each figure presented in text we derived 95% confidence intervals around the model coefficients." We did try higher iterations for bootstrapping but found no difference in model estimates or confidence intervals so present those conducted at 500). The figure headings of table S1-S5 have been updated to clarify that the models reported in text and in the figures are the interaction between land-use and the traits.

Reviewer #2 (Remarks to the Author):

Increasing pollination limitation is the expected consequence of pollinator decline, a by-product of man-driven conversion of natural habitats into extensive urban habitats and other types of land uses. The present MS "Land use and pollinator dependency drives global patterns of pollen limitation in the Anthropocene" addresses the question of whether land use intensification is associated with increasing pollen limitation by using a phylogenetically controlled meta-analysis based on a global dataset of over 2200 experimental studies and about 1200 wild plants. They found that urbanization as well as specialization increases pollination limitation.

Overall, this is an important, well-written and easy-to-read paper. I am quite certain that this study will be widely cited and have a high impact among policy makers and the general public. My main concerns are related to the model used for data analysis, as can be seen in the specific comments below. Also, I would like to see an extended discussion to explain the absence of a difference in pollen limitation between natural and managed land. This is particularly puzzling given the finding that habitat fragmentation frequently leads to higher pollination limitation (Aguilar et al. 2006). This could relate to the fact that plants thriving in managed land have slightly different profiles in terms of pollination and mating systems than those surviving in the fragments, as it implied in the MS (lines 114-117). Other possibilities worth mentioning are that heterogenous managed landscapes can be pollinator rich (Winfree et al. 2007), or plant-pollinator interactions can be quite resilience due to their asymmetric nature (Ashworth et al. 2004). Also, it is interesting that specialized plants are the most vulnerable independent of habitat type, which is quite surprising as I might expect increasing pollen limitation in the most disturbed habitats. Beyond these issues, I am very positive about this article and with some extra clarification can be a very suitable contribution for Nature Communications.

Response: We have made many improvements with regards to our model reporting in response to both reviewers' suggestions, including new tables which show the relative importance of all random effects (as explained above). We have also clarified the methods used to calculate the pooled variance in LN 11-15. We agree with the reviewer that plant communities that dominate managed verses natural habitats are likely to have species with different pollinators and mating systems and this is why we included a species-level phylogeny in our analysis. We acknowledge that previously it was not clear that our phylogeny was at the species level - an issue we have now clarified.

The reviewer also raises some interesting points about why plant species in natural and managed landscapes may be similarly pollen limited and we have updated the main text to incorporate these useful insights and to incorporate the suggested references in LN 118-122 of the main text: "For example, although many stressors associated with managed landscapes are known to lead to higher pollination limitation¹⁴, heterogeneously managed landscapes can increase pollinator diversity and therefore could lower pollen limitation¹⁰. Furthermore, the asymmetric nature of plant-pollinator interactions may make them resilient to some disturbance²⁸."

Specific comments

Line 97-99. It should be interesting to know the magnitude of phylogenetic patterning (e.g., Bloomberg's K) existing in the data set.

Response: We agree and have added this insight into LN 97- 98 of the main text, which now reads "We did not find significant phylogeny signal in PL in our highly geographically and species diverse dataset (K = 0.31, P = 0.097)."

Supplementary material

“We computed a single estimate of the magnitude of PL and its variance for each experiment (i.e., species, population, and year of study).” - PL is a transformed variable based on two input variables. The estimation of the variance of PL then needs clarification.

Response: We used standard equations for pooling variance in meta-analysis. Specifically, the straight forward equation outlined in Borenstein et al. 2011 and a more complex equation that outlines how to apply variance pooling across multiple cases/rows from Koricheva et al 2013. Thank you for noticing this omission, we have updated the methods text in LN 11-15, which now reads “In simple cases, a pooled variance was calculated following Ref⁴ i.e. when a row related to a single species population and year. In cases when data for a single species population were presented across multiple rows i.e. multiple time-periods (e.g., season) or multiple morphs (e.g., flower color, gender) variance was calculated following Ref⁵.”

“Plants were scored as ‘one’ when pollinated by one pollinator species, ‘few’ when pollinated by a few (2 to 4) species or ‘many’ (5 or more) pollinator species following.” - Most likely many studies report flower visitors rather than efficient pollinators. Therefore, clarification is needed here.

Response: This statement has been clarified in the methods text LN 53-55, which now reads “The degree of ecological specialization was based on the total number of known pollinators for the plant or the number of recorded flower visitors to the plant.”

“All analyses were performed in R8. We conducted phylogenetic mixed effects meta-analyses as per methods in5,9 with PL as the response variable and land use, levels of dependence on pollinators and their interaction as predictors”- What about the other fixed effects such as taxonomic and functional generalization? Also, there were more PL estimates than number of species. The presence of more than one PL estimate for some species needs clarification from a statistical point of views. Were estimates for single species averaged, or “species” included as a random factor?

Response: We have revised the methods text LN 69-73 to improve clarity, which now reads “We conducted phylogenetic mixed-effects meta-analyses as per methods in7,11 with PL as the response variable and the interaction between land use, and three plant traits that relate to their levels of dependence on pollinators (pollinator dependency, and ecological and functional specialisation on pollinators)”.

Yes, there are multiple estimates for some species, which makes our test more powerful when comparing among habitat types. The species level phylogeny is included as a random effect to account for inclusion of multiple estimates for the same species from different places. We realise that it was not clear and based on both reviewers comments we have clarified the methods text in LN 75-79, “The species level phylogeny used in our analysis is available on-line as part of the GloPL database¹ and is based on the published dated molecular phylogeny by Zanne and colleagues¹³”.

“Phylogeny was modelled as a variance-covariance matrix, which assumes Brownian motion like evolution, using the vcv function in the ape package¹² and was included as a random

effect in all models.” - Phylogeny is an explanatory variable of seemingly random variation among species. I wouldn't consider phylogeny to be a random effect.

“Because differences in experimental design affect the estimated magnitude of PL, for a review of their effects see¹³, we included in each model a random effect to control for differences in the response variables measured (fruit-set, seed-set, seeds per plant, seeds per flower, seeds per fruit), the level at which the treatment was applied (whole plant, partial plant, flower) and whether or not bags were applied to the plants.” - These are typical categories of fixed factors (i.e., “reproductive variable measured”, “treatment level”, and “bagging”); I am quite uncertain why they were considered random effects. Some clarification is needed.

Response: In this study our main aim is to broadly compare the level of pollen limitation among habitat types independent of experiment type or species composition between habitat types. Therefore, we included the experimental protocol and phylogeny as random effects. We have updated the text to clarify that the phylogeny was a species level phylogeny and used in part because we have multiple estimates from difference locations for some species. We have now included new tables S6-S11, which show the variance estimates from all random effects. They show that the phylogeny is highly important, much more so than experimental design. As there was no phylogenetic signal in PL this shows that there is phylogenetic signal in the explanatory variables (i.e. traits and land-use) and supports the need to include this random effect.

SOURCE DATA: In an effort to ensure reproducibility of research data, we now also require that you provide a separate source data file. The source data file should, as a minimum, contain the raw data underlying all reported averages in graphs. To learn more about our motivation behind this policy, please see <https://www.nature.com/articles/s41467-018-06012-8>.

Within the source data file, each figure or table (in the main manuscript and in the Supplementary Information) containing relevant data should be represented by a single sheet in an Excel document, or a single .txt file or other file type in a zipped folder. Blot and gel images should be pasted in and labelled with the relevant panel and identifying information such as the antibody used. We also encourage you to include any other types of raw data that may be appropriate. An example source data file is available demonstrating the correct format:

<https://www.nature.com/documents/ncomms-example-source-data.xlsx>

The file should be labelled ‘Source Data’, with the title and a brief description included in your cover letter, and should be mentioned in all relevant figure legends using the template text below:

“Source data are provided as a Source Data file.”

Reviewers' Comments:

Reviewer #2:

Remarks to the Author:

I read the revision of the MS "Land use and pollinator dependency drives global patterns of pollen limitation in the Anthropocene", and I am mostly happy with how my main concerns were addressed.

However, although the MS reads well it will benefit from careful editing. While reading it, I found several mistakes (particularly in the Methods section).

Here I list a few:

Line 81. I suggest to replace "should not have increased" by "should not increase".

Methods

Line 8. I suggest to replace ". Where" by ", where".

Lines 15. The first part of the sentence is missing.

Line 89. "t2" shouldn't be "Tau2"?

Line 93. "conduct" not "conducted"

Overall, I enjoyed very much reading this useful and significant article.

Reviewer #3:

Remarks to the Author:

I have been asked to assess whether the questions raised by Reviewer 1 have been adequately addressed. My comments are below the answers of the authors.

Reviewers' comments:

Reviewer #1 (Remarks to the Author):

My background is in meta-analysis, and I was asked to evaluate what the authors achieved in this manuscript. I like what they did, but some details are underreported, and below I outline these to improve the clarity of their meta-analysis.

Response: We appreciate the endorsement and thankful for the advice on additional reporting metrics. We now provide a k for each subgroup in analysis in tables S1, S3 and S4. We have also included new tables S6-S11 which provide the estimate for each random effect included in the model.

1) More details on the effect size is needed (see Suppl. Page 1). Were both the numerator and denominator means (i.e. averages of supplemented and natural)? Please include the equation used to estimate the variance of the effect size (e.g., $\ln RR$). A clear description of the number of effect sizes included in the meta-analysis is missing, as well as the number of effects included in each subgroup analyses. The sample sizes of meta-analysis is defined with k and not N (since N is used to define the sample size used to estimate individual effect sizes). Please report all k (number of effect sizes) for each pooled effect size reported in the MS.

Response: Yes, both the supplemented and natural i.e. the numerator and denominator are means

and we have revised the methods text on LN 4 to clarify this "Each experiment compared the mean reproductive output of plants receiving supplemental pollination applied by hand with those receiving natural pollination."

Response: The sample size (k) is now reported for each subgroup in the analysis. Revisions made to tables S1, S3 & S4.

All the concerns have been correctly addressed except one: "Please include the equation used to estimate the variance of the effect size (e.g., $\ln RR$)."

Instead of including the equation, the authors cite two books of >400 pages where the method is hard to find. More explanations should be given.

2) More information about the phylogeny used in modelling the meta-analysis is needed. Is this a species-level phylogeny? How were polytomies resolved? Surely not all species in the GloPI database were represented in the Zanne et al. phylogeny. Were species surrogates used for placement in the phylogeny? Perhaps all of this is all outlined in the GloPI publication, but some information is still needed here to make sense of the scope of this meta-analysis.

Response: Yes, the phylogeny is a species-level one and it was necessary to add some of our focal species to the tree. The methods text in LN 75 – 79 has been updated to clarify this point and now reads "The species-level phylogeny used in our analysis is available on-line as part of the GloPL database¹ and is based on the published dated molecular phylogeny by Zanne and colleagues¹¹. Focal species not present in the Zanne et al.¹³ phylogeny were bound on the tree when their genus was present, when not they were grafted on based on published information following methods outlined in GloPL."

This information clearly addresses the referee's questions.

3) There are many random-effects included in the meta-analysis. The variances from these random-effects can swamp the weighting of effects, and therefore at minimum, all of the random-effect variances should be reported for at least the overall grand-mean synthesis (reporting this in the supplements is ok). This is important so that the reader can assess whether conventional between-study variance (τ -squared) assumed by random-effects meta-analysis is washed out by the numerous phylogenetic, response type, and experimental design random factors. It is also not clear whether the conventional τ -square was included in the meta-analysis; this is a critical random-effect that needs to be included in all models and underlies fundamental principles of meta-analysis. This is not included as a default when the `rma.mv` function is used in metafor. To include it, it is simply a column of unique numbers for each effect sizes used in the model.

Response: We have now included new tables S6-S11, which show the variance estimates from all random effects. We have also re-run the analysis to include a random effect for each estimate (τ^2). Including a τ^2 random effect did not change our main result that urban environments are more pollen limited and none of the patterns previously observed have changed.

The new information is OK but τ^2 is missing from Tables S8 and S10

4) About the meta-analysis model itself, what method was used to estimate the multiple random-effects (e.g., REML, ML)? Why was bootstrapping used? This seems unnecessary as the reported metafor coefficients and their 95%CI are sufficient (also 500 repetitions is very low given the number of effects included). This style of bootstrapping is also not part of the standard metafor functions of analyses; permutation tests are applied in separate functions, were these used? How were the marginal means of each group estimated when multiple predictors are included in models (e.g., results reported in Figures 1 and 2)? The estimated error type is not reported in either figure. It's not clear if interactions were included in models (e.g., land use type + pollinator dependence + land use type * pollinator dependence), or why it was excluded.

Response: All models were fitted using REML, this has been clarified in the methods text LN 93 which now reads "All models were fit using REML." Bootstrapping was used for the figures and not for model interpretation, this has now been clarified in the methods text L97-99, which reads "For each figure presented in text we derived 95% confidence intervals around the model coefficients." We did try

higher iterations for bootstrapping but found no difference in model estimates or confidence intervals so present those conducted at 500). The figure headings of table S1-S5 have been updated to clarify that the models reported in text and in the figures are the interaction between land-use and the traits.

Although it is clear that bootstrapping was only used for figures, the justification on why 95% CI reported by metaphor were not used in the figures is still lacking. The question on "How were the marginal means of each group estimated when multiple predictors are included in models (e.g., results reported in Figures 1 and 2)?" remains unanswered.

Reviewer #4:

Remarks to the Author:

The results of the study by Bennett and co-authors make a timely and useful contribution to our understanding of the impacts of land use and land use intensification on pollen limitation of plant reproduction, particularly since they provide some insight into the spatial patterns of the impacts and how these may be related to functional and ecological traits as well as anthropogenic effects. The authors have already expanded the discussion of some of the findings, such as the absence of a difference in pollen limitation between natural and managed land. However, while most of the expanded text is understandable to the non-expert, the statement in lines 22-23 about asymmetry in plant-pollinator interactions will remain obscure to many readers. My main concern about the current version of the manuscript, noted in comments below, is the lack of detail or transparency about the choice of experiments to be included in the meta-analysis and about how and which species were added to the phylogeny of Zanne et al.

Main text

- While the text mostly was clear, the number of typos and grammatical errors I observed suggests that a careful check of the text appears not to have been made before submission. Some of these are highlighted in specific comments.

Methods

- I have not conducted the kind of meta-analysis described in this paper and so cannot comment on the appropriateness of specific details such which variance to use in different cases (lines 11-15 of Methods), but the basic details of the experimental design (lines 3-20 of Methods) were clear and easy to follow.

- Lines 3-20. However, it wasn't clear how the experiments to be included in the analysis were chosen. The 2247 experiments on 1247/1249 (not clear which number is correct; one number is used in line 88 of main text, another one in this section of the methods) do not represent the totality of experiments in the GloPL database. What made some more suitable for inclusion than others? I'm assuming there were good reasons, e.g., perhaps spatial coordinates were missing, but this should be spelled out.

- Missing word/s in sentence beginning on line 15 of Methods.

- Line 39 of Methods: "board" should be "broad".

- Line 52. Grammar.

- Line 64. "Direct" should be "direction".

- Lines 64-65. Fix punctuation or split into two sentences.

- Lines 76-78. There is inadequate detail to explain how species were added to the Zanne et al.

phylogeny. These details are also lacking from the GLoPL paper. For example, I work with phylogenetic trees all the time, but I do not know what it means when the authors say that the species "were bound to the tree when their species was present". Does this mean that a species in a genus was attached to a single branch leading to the relevant genus? Or does it mean that the species was placed on a branch within the genus? Or that it was added in a polytomy at the base of the genus clade? Similarly, I'm trying to understand how species in genera not in the Zanne et al. phylogeny were added to the tree. I can make some guesses about how this was done from the text in the GLoPL paper, but I have no idea if any of my guesses would be correct. Additionally, there is no information about how many species had to be added to the tree by hand, nor what traits they might represent. I imagine it is a small percentage, but the methods should be transparent here. If it is a small percentage then even if there is some bias in their ecological and/or functional traits, there might not be much impact on the outcome.

- Lines 85-86. Do you mean a priori? If so, correct.

Figures

- Figure 2a is very information-rich, but not very accessible. It may just be that it is too small.

Reviewer #2 (Remarks to the Author):

I read the revision of the MS “Land use and pollinator dependency drives global patterns of pollen limitation in the Anthropocene”, and I am mostly happy with how my main concerns were addressed.

However, although the MS reads well it will benefit from careful editing. While reading it, I found several mistakes (particularly in the Methods section).

RE: The manuscript and the supplementary material have received a thorough proofread. We have now fixed a number of typos, and we thank the reviewer for bringing this to our attention.

Here I list a few:

Line 81. I suggest to replace “should not have increased” by “should not increase”.

RE: Suggested edit made

Methods

Line 8. I suggest to replace “. Where” by “, where”.

RE: Suggested edit made

Lines 15. The first part of the sentence is missing.

RE: We have added the beginning of this sentence LN 205 now reads “We compared results with..”

Line 89. “t²” shouldn’t be “Tau²”?

RE: Suggested edit made

Line 93. “conduct” not “conducted”

RE: Suggested edit made

Overall, I enjoyed very much reading this useful and significant article.

RE: Thank you

Reviewer #3 (Remarks to the Author):

I have been asked to assess whether the questions raised by Reviewer 1 have been adequately addressed. My comments are below the answers of the authors.

Reviewers' comments:

Reviewer #1 (Remarks to the Author):

My background is in meta-analysis, and I was asked to evaluate what the authors achieved in this manuscript. I like what they did, but some details are underreported, and below I outline these to improve the clarity of their meta-analysis.

Response: We appreciate the endorsement and thankful for the advice on additional reporting metrics. We now provide a k for each subgroup in analysis in tables S1, S3 and S4. We have also included new tables S6-S11 which provide the estimate for each random effect included in the model.

1) More details on the effect size is needed (see Suppl. Page 1). Were both the numerator and denominator means (i.e. averages of supplemented and natural)? Please include the equation used to estimate the variance of the effect size (e.g., $\ln RR$). A clear description of the number of effect sizes included in the meta-analysis is missing, as well as the number of effects included in each subgroup analyses. The sample sizes of meta-analysis is defined with k and not N (since N is used to define the sample size used to estimate individual effect sizes). Please report all k (number of effect sizes) for each pooled effect size reported in the MS.

Response: Yes, both the supplemented and natural i.e. the numerator and denominator are means and we have revised the methods text on LN 4 to clarify this “Each experiment compared the mean reproductive output of plants receiving supplemental pollination applied by hand with those receiving natural pollination.”

Response: The sample size (k) is now reported for each subgroup in the analysis. Revisions made to tables S1, S3 & S4.

All the concerns have been correctly addressed except one: “Please include the equation used to estimate the variance of the effect size (e.g., $\ln RR$).” Instead of including the equation, the authors cite two books of >400 pages where the method is hard to find. More explanations should be given.

RE: To calculate pooled variances when there were multiple entries for a single population we used Borenstein (Ref 37 in manuscript) formulae 11.2 to 11.4 pages 65-66. For simple cases (i.e. the row in GloPL relates to a single population), pooled variance was calculated using Koricheva, Handbook of Meta-analysis in Ecology and Evolution, p64 (Ref 36 in manuscript), which is the same one used in Hedges, L. V., Gurevitch, J., & Curtis, P. S. (1999). The meta-analysis of response ratios in experimental ecology. Ecology, 80(4), 1150-1156. We have updated the text in LN 200 and 203 to include page and formula numbers.

2) More information about the phylogeny used in modelling the meta-analysis is needed. Is this a species-level phylogeny? How were polytomies resolved? Surely not all species in the GloPI database were represented in the Zanne et al. phylogeny. Were species surrogates used for placement in the phylogeny? Perhaps all of this is all outlined in the GloPI publication, but some information is still needed here to make sense of the scope of this meta-analysis.

Response: Yes, the phylogeny is a species-level one and it was necessary to add some of our focal species to the tree. The methods text in LN 75 – 79 has been updated to clarify this point and now reads “The species-level phylogeny used in our analysis is available on-line as part of the GloPL database¹ and is based on the published dated molecular phylogeny by Zanne and colleagues¹¹. Focal species not present in the Zanne et al.¹³ phylogeny were bound on the tree when their genus was present, when not they were grafted on based on published information following methods outlined in GloPL.”

This information clearly addresses the referee's questions.

RE: Thank you

3) There are many random-effects included in the meta-analysis. The variances from these random-effects can swamp the weighting of effects, and therefore at minimum, all of the random-effect variances should be reported for at least the overall grand-mean synthesis (reporting this in the supplements is ok). This is important so that the reader can assess whether conventional between-study variance (τ -squared) assumed by random-effects meta-analysis is washed out by the numerous phylogenetic, response type, and experimental design random factors. It is also not clear whether the conventional τ -square was included in the meta-analysis; this is a critical random-effect that needs to be included in all models and underlies fundamental principles of meta-analysis. This is not included as a default when the `rma.mv` function is used in `metafor`. To include it, it is simply a column of unique numbers for each effect sizes used in the model.

Response: We have now included new tables S6-S11, which show the variance estimates from all random effects. We have also re-run the analysis to include a random effect for each estimate (τ^2). Including a τ^2 random effect did not change our main result that urban environments are more pollen limited and none of the patterns previously observed have changed.

The new information is OK but τ^2 is missing from Tables S8 and S10

RE: For comparison, the results presented in table S8 and S10 are from models without τ^2 and Table S7 and S9 are from models with τ^2 . The table captions for S8 and S10 now included the statement "without τ^2 " to clarify this.

4) About the meta-analysis model itself, what method was used to estimate the multiple random-effects (e.g., REML, ML)? Why was bootstrapping used? This seems unnecessary as the reported `metafor` coefficients and their 95%CI are sufficient (also 500 repetitions is very low given the number of effects included). This style of bootstrapping is also not part of the standard `metafor` functions of analyses; permutation tests are applied in separate functions, were these used? How were the marginal means of each group estimated when multiple predictors are included in models (e.g., results reported in Figures 1 and 2)? The estimated error type is not reported in either figure. It's not clear if interactions were included in models (e.g., land use type + pollinator dependence + land use type * pollinator dependence), or why it was excluded.

Response: All models were fitted using REML, this has been clarified in the methods text LN 93 which now reads "All models were fit using REML." Bootstrapping was used for the figures and not for model interpretation, this has now been clarified in the methods text L97-99, which reads "For each figure presented in text we derived 95% confidence intervals around the model coefficients." We did try higher iterations for bootstrapping but found no difference in model estimates or confidence intervals so present those conducted at 500). The figure headings of table S1-S5 have been updated to clarify that the models reported in text and in the figures are the interaction between land-use and the traits.

Although it is clear that bootstrapping was only used for figures, the justification on why 95% CI reported by `metafor` were not used in the figures is still lacking.

The question on "How were the marginal means of each group estimated when multiple predictors

are included in models (e.g., results reported in Figures 1 and 2)?" remains unanswered.

RE: The metaphor package reports CI-s based on SE and z-value, which assume symmetrical CI-s around a mean and a normal distribution. The 95% bootstrap CI-s do not assume any kind of distribution for the coefficients of the model (Hesterberg, 2015). Thus, bootstrapped CI-s are more robust than reporting the CI-s directly from the model based on an assumption of normality and give more realistic CI-s values which are better for visualising the differences between groups.

Hesterberg, Tim C. "What teachers should know about the bootstrap: Resampling in the undergraduate statistics curriculum." *The American Statistician* 69.4 (2015): 371-386.

Marginal means in the bootstrapped models were extracted by running models without the intercept in the metaphor package. Text in LN 303 has been updated to better describe how the marginal means in the bootstrapped models were extracted "Marginal means for each group present in figure 1 were extracted by running bootstrapped models fit with ML without the intercept." Bootstrap models were run using ML and REML and the figures presented in text were generated from the ML fit models. We tested for differences in coefficients values between employing ML and REML methods and found the differences were very small and did not our results. However, in text and in tables we presented results from REML although this inconsistency has no effect on the results, for consistency we now present models fit using ML throughout the MS and have updated the text and supplementary tables accordingly. We have also updated text in LN 292, which now reads ". All models presented here were fit using ML no quantitative differences were detected when compared with models fit using REML."

Reviewer #4 (Remarks to the Author):

The results of the study by Bennett and co-authors make a timely and useful contribution to our understanding of the impacts of land use and land use intensification on pollen limitation of plant reproduction, particularly since they provide some insight into the spatial patterns of the impacts and how these may be related to functional and ecological traits as well as anthropogenic effects. The authors have already expanded the discussion of some of the findings, such as the absence of a difference in pollen limitation between natural and managed land. However, while most of the expanded text is understandable to the non-expert, the statement in lines 22-23 about asymmetry in plant-pollinator interactions will remain obscure to many readers. My main concern about the current version of the manuscript, noted in comments below, is the lack of detail or transparency about the choice of experiments to be included in the meta-analysis and about how and which species were added to the phylogeny of Zanne et al.

RE: We thank the reviewer for their comments. We now provide an explanation of asymmetry in plant-pollinator networks in LN 126 "Furthermore, the asymmetric nature of plant-pollinator interactions, where specialist plant species are often pollinated by generalist pollinators may make them resilient to some disturbance²⁸". We have provided more details on the analysis and phylogeny as detailed below.

Main text

- While the text mostly was clear, the number of typos and grammatical errors I observed suggests

that a careful check of the text appears not to have been made before submission. Some of these are highlighted in specific comments.

RE: The manuscript and the supplementary material have received a thorough proofread. We have now fixed a number of typos, and we thank the reviewer for bring this to our attention.

Methods

- I have not conducted the kind of meta-analysis described in this paper and so cannot comment on the appropriateness of specific details such which variance to use in different cases (lines 11-15 of Methods), but the basic details of the experimental design (lines 3-20 of Methods) were clear and easy to follow.

RE: We thank the reviewer for their comments.

- Lines 3-20. However, it wasn't clear how the experiments to be included in the analysis were chosen. The 2247 experiments on 1247/1249 (not clear which number is correct; one number is used in line 88 of main text, another one in this section of the methods) do not represent the totality of experiments in the GloPL database. What made some more suitable for inclusion than others? I'm assuming there were good reasons, e.g., perhaps spatial coordinates were missing, but this should be spelled out.

RE: We used all populations under natural conditions in the global dataset. Summary statistics such as the mean global level of PL and phylogenetic signal in PL are conducted on all 2247 natural populations in GLOPL. However, the reviewer is correct that the models can only be performed on populations with data on all fixed and random effects. This means only populations with geospatial coordinates, data on experimental design and where traits for pollinator dependency are known – this was the case for 2198 studies when using the PL effect size without 0.5 and explains the disparity between K in the models and the summaries provided in text. To make this clearer we have updated the text in LN 305, which now reads “For all natural populations in GloPL with geographic coordinates, data on all random effect and with known pollinator dependency were included in modelled analysis”. We have revised LN 100 to indicate when all cases in the GLoPI dataset were used to generate summary statistics.

- Missing word/s in sentence beginning on line 15 of Methods.

RE: We have added the beginning of this sentence LN 205 now reads “We compared results with..”

- Line 39 of Methods: "board" should be "broad".

RE: Suggested edit made

- Line 52. Grammar.

Re: The original LN 52 has been revised for clarity LN 243 now reads “Levels of pollination specialization was scored based on the authors determination in the original studies”.

- Line 64. "Direct" should be "direction".

RE: Suggested edit made

- Lines 64-65. Fix punctuation or split into two sentences.

RE: The sentence has been split into two in LN254-257.

- Lines 76-78. There is inadequate detail to explain how species were added to the Zanne et al. phylogeny. These details are also lacking from the GLoPL paper. For example, I work with phylogenetic trees all the time, but I do not know what it means when the authors say that the species "were bound to the tree when their species was present". Does this mean that a species in a genus was attached to a single branch leading to the relevant genus? Or does it mean that the species was placed on a branch within the genus? Or that it was added in a polytomy at the base of the genus clade? Similarly, I'm trying to understand how species in genera not in the Zanne et al. phylogeny were added to the tree. I can make some guesses about how this was done from the text in the GLoPL paper, but I have no idea if any of my guesses would be correct. Additionally, there is no information about how many species had to be added to the tree by hand, nor what traits they might represent. I imagine it is a small percentage, but the methods should be transparent here. If it is a small percentage then even if there is some bias in their ecological and/or functional traits, there might not be much impact on the outcome.

RE: We now provide additional information to more clearly outline how many species were missing from the Zanne et al supertree and how they were added. LN 268-276 now reads "To create the phylogeny, we started with the dated supertree created by Zanne and colleagues⁴¹. Species that were not included in the supertree, were bound to the tree when their genus was present by creating a polytomy with congeners that were present in the tree using the `congeneric.merge` function from the "pez" package in R⁴². When no congener was present, as was the case for 60 of the GLoPL species, we searched the literature for published phylogenies indicating closely-related genera and manually grafted these species to their sister genera (or tribes). We then pruned the supertree to only include our focal species using the `drop.tip` function from the "ape" package⁴³"

- Lines 85-86. Do you mean a priori? If so, correct.

RE: Suggested revision made

Figures

- Figure 2a is very information-rich, but not very accessible. It may just be that it is too small.

RE: Figure 2 has been enlarged and a high definition pdf has also been uploaded.

Reviewers' Comments:

Reviewer #4:

Remarks to the Author:

I was asked to evaluate the reviewers responses to my comments. I feel that my concerns have been addressed and appreciate the clarifying detail that was provided. Just two points below:

- Lines 3-20. However, it wasn't clear how the experiments to be included in the analysis were chosen. The 2247 experiments on 1247/1249 (not clear which number is correct; one number is used in line 88 of main text, another one in this section of the methods) do not represent the totality of experiments in the GloPL database. What made some more suitable for inclusion than others? I'm assuming there were good reasons, e.g., perhaps spatial coordinates were missing, but this should be spelled out.

RE: We used all populations under natural conditions in the global dataset. Summary statistics such as the mean global level of PL and phylogenetic signal in PL are conducted on all 2247 natural populations in GLOPL. However, the reviewer is correct that the models can only be performed on populations with data on all fixed and random effects. This means only populations with geospatial coordinates, data on experimental design and where traits for pollinator dependency are known – this was the case for 2198 studies when using the PL effect size without 0.5 and explains the disparity between K in the models and the summaries provided in text. To make this clearer we have updated the text in LN 305, which now reads "For all natural populations in GloPL with geographic coordinates, data on all random effect and with known pollinator dependency were included in modelled analysis". We have revised LN 100 to indicate when all cases in the GLoPI dataset were used to generate summary statistics.

Reviewer #4 Response. The line numbers provided above are misleading (e.g., LN 305 is in the references), but the clarification and increased transparency is helpful.

- Line 52. Grammar.

Re: The original LN 52 has been revised for clarity LN 243 now reads "Levels of pollination specialization was scored based on the authors determination in the original studies".

Reviewer #4 Response: Great, but now "was" should be "were" in order to agree with plural "levels".

- Lines 76-78. There is inadequate detail to explain how species were added to the Zanne et al. phylogeny. These details are also lacking from the GLoPL paper. For example, I work with phylogenetic trees all the time, but I do not know what it means when the authors say that the species "were bound to the tree when their species was present". Does this mean that a species in a genus was attached to a single branch leading to the relevant genus? Or does it mean that the species was placed on a branch within the genus? Or that it was added in a polytomy at the base of the genus clade? Similarly, I'm trying to understand how species in genera not in the Zanne et al. phylogeny were added to the tree. I can make some guesses about how this was done from the text in the GLoPL paper, but I have no idea if any of my guesses would be correct. Additionally, there is no information about how many species had to be added to the tree by hand, nor what traits they might represent. I imagine it is a small percentage, but the methods should be transparent here. If it is a small percentage then even if there is some bias in their ecological and/or functional traits, there might not be much impact on the outcome.

RE: We now provide additional information to more clearly outline how many species were missing from the Zanne et al supertree and how they were added. LN 268-276 now reads "To create the phylogeny, we started with the dated supertree created by Zanne and colleagues⁴¹. Species that were not included in the supertree, were bound to the tree when their genus was present by creating a polytomy with congeners that were present in the tree using the `congeneric.merge` function from the

"pez" package in R42. When no congener was present, as was the case for 60 of the GloPL species, we searched the literature for published phylogenies indicating closely-related genera and manually grafted these species to their sister genera (or tribes). We then pruned the supertree to only include our focal species using the drop.tip function from the "ape" package⁴³

Reviewer #4 Response: Very helpful. And I assume by "manually grafting" you mean that you attached the species being grafted onto the tree to the branch leading to the genus clade (if there was more than one species).

REVIEWERS' COMMENTS:

Reviewer #4 (Remarks to the Author):

I was asked to evaluate the reviewers responses to my comments. I feel that my concerns have been addressed and appreciate the clarifying detail that was provided. Just two points below:

- Lines 3-20. However, it wasn't clear how the experiments to be included in the analysis were chosen. The 2247 experiments on 1247/1249 (not clear which number is correct; one number is used in line 88 of main text, another one in this section of the methods) do not represent the totality of experiments in the GloPL database. What made some more suitable for inclusion than others? I'm assuming there were good reasons, e.g., perhaps spatial coordinates were missing, but this should be spelled out.

RE: We used all populations under natural conditions in the global dataset. Summary statistics such as the mean global level of PL and phylogenetic signal in PL are conducted on all 2247 natural populations in GLOPL. However, the reviewer is correct that the models can only be performed on populations with data on all fixed and random effects. This means only populations with geospatial coordinates, data on experimental design and where traits for pollinator dependency are known – this was the case for 2198 studies when using the PL effect size without 0.5 and explains the disparity between K in the models and the summaries provided in text. To make this clearer we have updated the text in LN 305, which now reads “For all natural populations in GloPL with geographic coordinates, data on all random effect and with known pollinator dependency were included in modelled analysis”. We have revised LN 100 to indicate when all cases in the GLOPI dataset were used to generate summary statistics.

Reviewer #4 Response. The line numbers provided above are misleading (e.g., LN 305 is in the references), but the clarification and increased transparency is helpful.

Response: Thank you

- Line 52. Grammar.

Re: The original LN 52 has been revised for clarity LN 243 now reads “Levels of pollination specialization was scored based on the authors determination in the original studies”.

Reviewer #4 Response: Great, but now “was” should be “were” in order to agree with plural “levels”.

Response: Suggested revision made.

- Lines 76-78. There is inadequate detail to explain how species were added to the Zanne et al. phylogeny. These details are also lacking from the GLOPL paper. For example, I work with phylogenetic trees all the time, but I do not know what it means when the authors say that the species "were bound to the tree when their species was present". Does this mean that a species in a genus was attached to a single branch leading to the relevant genus? Or does it mean that the species was placed on a branch within the genus? Or that it was added in a polytomy at the base of the genus clade? Similarly, I'm trying to understand how species in genera not in the Zanne et al. phylogeny were added to the tree. I can make some guesses about how this was done from the text

in the GLoPL paper, but I have no idea if any of my guesses would be correct. Additionally, there is no information about how many species had to be added to the tree by hand, nor what traits they might represent. I imagine it is a small percentage, but the methods should be transparent here. If it is a small percentage then even if there is some bias in their ecological and/or functional traits, there might not be much impact on the outcome.

RE: We now provide additional information to more clearly outline how many species were missing from the Zanne et al supertree and how they were added. LN 268-276 now reads "To create the phylogeny, we started with the dated supertree created by Zanne and colleagues⁴¹. Species that were not included in the supertree, were bound to the tree when their genus was present by creating a polytomy with congeners that were present in the tree using the `congeneric.merge` function from the "pez" package in R⁴². When no congener was present, as was the case for 60 of the GLoPL species, we searched the literature for published phylogenies indicating closely-related genera and manually grafted these species to their sister genera (or tribes). We then pruned the supertree to only include our focal species using the `drop.tip` function from the "ape" package⁴³"

Reviewer #4 Response: Very helpful. And I assume by "manually grafting" you mean that you attached the species being grafted onto the tree to the branch leading to the genus clade (if there was more than one species).

Response: Yes, we grafted to the branch leading to the genus clade. We have clarified this in LN 303.